# Birthing under the Condition of the COVID-19 Pandemic in Germany: Interviews with Mothers, Partners, and Obstetric Health Care Workers

**DOI:** 10.3390/ijerph19031486

**Published:** 2022-01-28

**Authors:** Martina Schmiedhofer, Christina Derksen, Johanna Elisa Dietl, Freya Häussler, Frank Louwen, Beate Hüner, Frank Reister, Reinhard Strametz, Sonia Lippke

**Affiliations:** 1German Coalition for Patient Safety (Aktionsbündnis Patientensicherheit), 10179 Berlin, Germany; j.dietl@jacobs-university.de (J.E.D.); haeussler@aps-ev.de (F.H.); strametz@aps-ev.de (R.S.); 2Department of Psychology & Methods, Jacobs University Bremen GmbH, 28759 Bremen, Germany; C.Derksen@jacobs-university.de (C.D.); S.Lippke@jacobs-university.de (S.L.); 3Department of Gynecology and Obstetrics, Division of Obstetrics and Prenatal Medicine, University Hospital Frankfurt, 60596 Frankfurt am Main, Germany; Frank.Louwen@kgu.de; 4Department of Gynecology and Obstetrics, University Hospital Ulm, 89070 Ulm, Germany; beate.huener@uniklinik-ulm.de (B.H.); frank.reister@uniklinik-ulm.de (F.R.); 5Wiesbaden Business School, Rhein Main University of Applied Science, 65183 Wiesbaden, Germany

**Keywords:** COVID-19 pandemic, obstetrics, birth, qualitative research, delivery, obstetric health care workers, HCW, patient safety, health care research

## Abstract

Background: The COVID-19 pandemic and the necessary containment measures challenge obstetric care. Support persons were excluded while protection measures burdened and disrupted the professionals’ ability to care and communicate. The objective of this study was to explore the first-hand experience of the impact of the COVID-19 pandemic on mothers, their partners, and obstetric professionals regarding birth and obstetric care in a university hospital. Methods: To answer the descriptive research questions, we conducted a qualitative content analysis using a data triangulation approach. We carried out 35 semi-structured interviews with two stratified purposive samples. Sample one consisted of 25 mothers who had given birth during the pandemic and five partners. Sample two included 10 obstetric professionals whose insights complemented the research findings and contributed to data validation. Participants were recruited from the study sample of a larger project on patient safety from two German university hospitals from February to August 2021. The study was approved by two ethics committees and informed consent was obtained. Results: Mothers complied with the rules, but felt socially isolated and insecure, especially before transfer to the delivery room. The staff equally reported burdens from their professional perspective: They tried to make up for the lack of partner and social contacts but could not live up to their usual professional standards. The exclusion of partners was seen critically, but necessary to contain the pandemic. The undisturbed time for bonding in the maternity ward was considered positive by both mothers and professionals. Conclusion: The negative effects of risk mitigation measures on childbirth are to be considered carefully when containment measures are applied.

## 1. Introduction

The COVID-19 pandemic has challenged all institutional and societal areas worldwide. To contain the pandemic, states have issued a variety of voluntary and enforceable risk mitigation measures including physical distancing and hygienic measures [1]. The German Federal State imposed an amendment of the Infection Protection Act that restricted social interaction as far as possible [2]. The main goal was to prevent overwhelming the healthcare systems by reducing elective procedures and stocking up on protective equipment [3].

Nevertheless, the health care system was hit largely unprepared. Pandemic plans were insufficient and personal protective equipment was lacking in the necessary quantities [4]. The COVID-19 pandemic emphasized existing structural problems regarding digitalization, training, and equipment as well as the lack of personnel resources [5]. Health care workers (HCW) had to restructure organizational processes and facilities to provide sufficient resources for COVID-19 patients. Hospitals introduced a no-visitation policy, that only allowed short-term visits for terminally ill patients or very young children. In-hospital care was limited to emergencies and urgent treatments, elective surgeries were canceled [6,7]. However, this was clearly not possible for pregnancies and births so HCW, as well as pregnant women and their partners, were forced to adapt to new and highly restrictive conditions, especially during the birth process itself [8,9]. The severity of these challenges becomes obvious when looking at global players: To counter the greater risk of unsafe childbirth due to the breakdown of health services caused by the COVID-19 pandemic, the WHO dedicated Patient Safety Day 2021 to ‘Safe maternal and newborn care’ [10]. The objective of the study’s problem is to understand the challenges better.

The pandemic affected the course of pregnancies and births in all areas: Women were alone at a doctor’s appointment and peer support was rare [11,12]. In the hospitals, women had elevated levels of anxiety due to a fear of infection. In particular, women who tested positive for COVID-19 perceived a lack of support or had a negative (birth) experience in the hospital [13]. The situation for the fathers-to-be was also challenging. Although their involvement and active participation in the birth process is seen as a central part of obstetric care, the pandemic restrictions kept them from such a role: Instead of joining and supporting the pregnant woman during consultations and the birth, they were forcibly left out [14,15].

Obstetricians, like all HCW, were faced with professional challenges but at the same time also faced challenges in their private lives [16]. They were not only double burdened but rather, the demands of each area increased the likelihood for those of the other: While they posed an additional risk of infection to their family members and close personal contacts, contagions spread from infected persons in the private environment could transmit the disease to the hospital and required quarantine measures [17].

At the same time, HCW had to advocate and uphold the regulations to their patients while providing support and had to also adapt their private lives to the containment rules [18]. In addition, it remained unclear for a long period of time which effects COVID-19 had on obstetric outcomes. A recent review unveiled that stillbirth, prematurity and low birth weight, asphyxia, and neonatal death were associated with neonatal outcomes of a mother’s COVID-19 status [19]. Pregnant women faced cesarean (C-)sections frequently due to different restraints and complications [20]. Caring for infected women medically and emotionally while preventing the spread of COVID-19 inside the hospital was perceived as challenging by HCW. Treating a widely unknown condition is associated with uncertainty and stress [21]. To deal with these restrictions, interdisciplinary collaboration has become even more important but remains challenging [22]. Thus, knowing the different factors is imperative.

### Objective

Against the background of this exceptional situation, we wanted to explore the impact of the COVID-19 pandemic on mothers, their partners, and obstetric professionals. To gain a deeper understanding of their perceptions, we chose a qualitative research design. We aimed to identify the perceived resilience and weaknesses of the obstetrics health care system in managing childbirth during this pandemic by answering the following research questions:A.What was the subjective impact of the restrictions to contain the COVID-19 pandemic on birth experiences from (becoming) mother’s and partner’s perspectives?
1.How did the COVID-19 regulations impact all phases of birth from the mother’s perspectives?a.During the first phase of birth?b.At the delivery room?c.At the maternity ward?2.How was the accompanying partner affected?3.How could the perceived burdens be managed?4.How did a COVID-19 diagnosis challenge the birth?B.What was the impact of COVID-19 pandemic restrictions on childbirth from the perspective of obstetric staff?
1.What adjustments did the COVID-19 pandemic containment rules require?2.How was the caring for the (pregnant) women affected?3.How were the professional obstetric standards affected?4.How did the mothers (and partners) cope with the restrictions from the professionals’ point of view?5.How could the birth of COVID-19 infected mothers be managed?C.What were resilience and weakness factors of obstetric health care under COVID-19 conditions?

## 2. Materials and Methods

The current study was part of the research project ‘TeamBaby’, an intervention study conducted at two German University hospitals. The purpose of ‘TeamBaby’ is to improve professionals’ and patients’ communication skills and thus to reduce preventable adverse events (pAE) [23]. Detailed information has been described during study registration in ClinicalTrials.org (NCT03855735) and in the published study protocol [24]. 

The objective of the present study is the investigation of the subjective impact of the recently spread COVID-19 pandemic on obstetrics. Since there is little knowledge so far, we decided on descriptive research questions. The data derive from qualitative semi-structured interviews, conducted with mothers, their partners, and obstetric HCW who were already familiar with the research project. 

### 2.1. Data Collection 

The maternal study participants and the partners were recruited from 219 mothers and 65 partners, who had attended the communication group training online prior to the birth. The professional study participants were selected out of 141 HCW who had taken part in the interprofessional trainings personally at a study clinic. To obtain data from respondents with differences in demographic characteristics, personal experiences, and different time periods of the pandemic, two stratified samples were built. Potential maternal respondents were approached according to their age, migration history, C-section or other medical complications (e.g., COVID-19 status), and time of birth. Professional participants were invited to the study based on their occupation, professional position, and length of work experience [25]. Both samples were enrolled equally from the study hospitals.

Between February and July 2021, one author (MS) carried out 25 semi-structured interviews with mothers and five partners (Table 1). There were 31 women, who were approached via email. The final number of interviews conducted was 25. Two declined the request for personal reasons and four did not answer the email. Ten complementary interviews with midwives, doctors and nurses were conducted in July and August 2021 (Table 2). Interviews were terminated as data saturation was reached and no new topics arose [26]. The open-ended interview guides (Table 3 and Table 4) included the key research issues which were flexibly adapted to the narrative flow and the individual openness of the participants. Most topics were reported and reflected in the course of the narrative, so few follow-up questions were needed.

### 2.2. Data Analysis

To answer the research questions, we used qualitative content analysis (QCA), which is a method suitable for descriptive qualitative data analysis [27]. Results are systematically worked out in a sequence of transparent and reliable steps by classifying verbatim parts of the data to predefined and emerging categories of a coding frame. The coding frame is created in a multi-stage process and flexibly adapted to emerging main dimensions and their subcategories. The research questions are answered in a concept-driven (inductive) and data-driven manner (deductive). QCA reduces the data to relevant aspects of the material which corresponds to the research concept of the study on hand. This approach focuses on birthing experiences under the COVID-19 pandemic and how they relate to perceptions of obstetric care. By comparing codes across persons, QCA provides intersubjectivity and thereby consistency and transferability.

Following each interview, field notes were taken to document impressions on atmosphere, nonverbal communication, quality of the internet connection with maternal respondents, and special features for confirmability. The interviews were audiotaped and transcribed verbatim by trained students of Jacobs University Bremen. All interview transcripts and field notes were entered into the qualitative data software MAXQDA2020 and anonymized for analysis. In the first stage, one of the authors (MS) reviewed the transcripts and coded them line by line. Then, 2030 sentence chunks or single words were labeled with a broad categorization. The data interpretation followed the maternal respondents’ narratives about their perceptions and experiences during all stages of birth from hospital admission to discharge. The same was carried out with the respective professional respondents’ narratives about the perceived change in providing care in their area of responsibility (e.g., delivery room or maternity ward). After several discussions with the research group (JD, FH, CD, consisting of psychologists, masters of public health, and one sociologist), 27 main categories (14 mothers/partners resp. 13 HCW) with 73 subcategories were built from similarities and contrasts within and between both of the data sources. Derived from the categories, common and distinct patterns from maternal, partners and professional respondents were used to answer the research questions. 

### 2.3. Triangulation

To obtain a comprehensive insight from different perspectives, we decided to perform the data triangulation method with obstetric HCW. By interviewing obstetricians, midwives, and nurses, we complemented the research findings with the expertise and perceptions of obstetric physicians, midwives and nurses and contributed to the validation of data [28]. HCW interviews were conducted subsequently after the interviews with mothers and partners. Maternal narratives were actively considered and professional views (e.g., organizational challenges) were explored in addition.

### 2.4. Ethical Considerations

The conduction of interviews about personal experiences in vulnerable situations requires careful ethical considerations. Even though childbirth is supposed to be a pleasant event, a recall of burdened experiences may cause emotional distress [29]. We, therefore, applied a strategy to protect the interviewees in our study concept. This relates mainly to the maternal participants, as the interviews with staff were conducted as expert interviews [30]. All participants were familiar with the goal of the TeamBaby project and had previously taken part in a safe communication training, which is the core of the ‘TeamBaby project’. The enrollment was performed in two steps. First, all interviewees were approached by research staff at the clinic sites, with whom all professional and maternal attendees of the communication trainings had already had personal contact. During the request, the objective of the study on hand was explained in detail, and questions were answered. If consent was given, one researcher (MS) invited maternal participants to online or phone interviews and professional participants to interviews conducted personally at the clinics. After the agreement, respondents received detailed written information prior to the interview by mail. To account for infants’ care needs respective working shifts, survey timing was flexibly scheduled by the participants. Before starting the data collection, all participants provided written informed consent. The interviewer emphasized that participation was voluntary and could be withdrawn at any time. Interview duration and thematic depth were determined by the interviewees. However, most maternal respondents and all of the HCW reported very openly. Some of the interviewed mothers had even reviewed the birth experience with their partners in preparation for the interview. This also resulted in the request of some partners to bring in their first-hand experience and actively take part in the interview. The collected data were used in a pseudonymized manner starting with the transcription. Codes were used for mothers (e.g., M_01) and HCW (e.g., HCW_01_Midwife) according to their study enrollment. All data were used only for research purposes that were clearly described in the prior information.

Ethical approval for data collection was granted through the project’s ethical approval from the University Hospital of Frankfurt Medical Research Ethics Committee (Number 19-292) and the University Hospital of Ulm Human Research Ethics Committee (Number 114/19). 

## 3. Results

This section begins with the main findings of the perceived impacts of the COVID-19 restrictions from mothers’ (and partners’) perspectives. Next, the results of the interviews with HCW are presented. The main findings are cross-tabulated in Table 5. 

Subsequently, we derived the perceived resilience and weaknesses of obstetrics in managing this pandemic from the data. An overview is given in Figure 1.

Data interpretation is based on interviewees’ narratives. The main topics correspond to the research questions and are presented in the continuous text. To highlight the results, and to provide transparency, key quotations are presented in text matrices. Each quote is labeled with the respective pseudonym, whereby M stands for maternal and HCW for professional participants. 

### 3.1. How Did the COVID-19 Regulations Impact All Phases of Birth and Pregnancy from the Mother’s and Partner’s Perspectives?

The maternal participants gave birth between July 2020 and May 2021, which covers a period with severe risk mitigation measures for women, including the partners’ exclusion from antenatal outpatient care, the delivery room, and maternity ward to the period with facilitated clinical access due to testing and vaccination availability. Furthermore, the sample includes women who experienced the outbreak unexpectedly during their pregnancy up to those who became pregnant during the pandemic. Accordingly, subjective perceptions were modified by the timing of the initial confrontation as well as by the level of restrictions. The presentation follows the stages of birth, with major differences reported between the periods before, during the time in the delivery room, and in the maternity ward (Table A1).

#### 3.1.1. During the First Phase of Birth

Some women’s accounts of their experiences during the first stage of labor are drastic and mirror the extent of the restrictions. As the partner was not allowed to join the woman inside the hospital, many participants attended the hospital as late as possible. They were entirely dependent on the hospital staff. However, clinical care in the early phase of labor was not as intensive as it is in the delivery room and the absence of accompanying persons could not be compensated for. Expectant mothers are supposed to stimulate labor contractions by walking around, climbing stairs, or coping with the labor contractions but were left on their own. Against that background, many interviewees said that they felt alone and missed social support and professional guidance before they were admitted to the delivery room (Table A1). 

#### 3.1.2. At the Delivery Room

The difference to non-pandemic times was the least during expulsion in the delivery room. Except for four weeks (when in one hospital no external person was allowed in), partners were permitted to physically accompany the expectant mother during that phase. Some interviewees mentioned how relieved and grateful they were to finally have their partner by their side. Furthermore, obstetric staff were, except for wearing face masks, on duty as in pre-pandemic times. Physical distancing had to be maintained for as long as medically feasible, but women in labor were allowed to remove face masks in the final phase of the birthing process. Respondents considered the rules appropriate but described negative effects on mutual understanding and emotional well-being.

To bond as a family, they were allowed to stay in the delivery room after the baby was born. Since the accompanying persons had to leave when the mother was transferred to the maternity ward together with the baby, this time was highly appreciated (Table A1).

#### 3.1.3. At the Maternity Ward 

From April 2020 to May 2021, visitors were not permitted inside the maternity ward. Since the rules were known in advance, most interviewees complied. However, the accompanying partner was missed and for some interviewees, it was upsetting to fully rely on the clinical staff. Nevertheless, apart from missing the partner, most women reported that they dealt with the separation well. Interestingly, other visitors were not at all or hardly missed. Some interviewees even valued the togetherness with the baby and would have been bothered by other visitors (including roommates) (Table A1). 

### 3.2. How Was the Accompanying Partner Affected?

The partners’ situation was reported from the mothers’ perspective and partners’ first-hand experience. The co-interviewed partners revealed helplessness as their predominant feeling during the period when they were not allowed inside the hospital. If partners could not commute home due to distance, they had to wait in the cold (in winter) or heat (in summer) in front of the hospital, without any possibility to buy food or beverages or access sanitary facilities. The hospitals, which used to be freely accessible, had turned into a kind of “fortress”, guarded by security services. The entrance to the delivery room had to sometimes be negotiated, which increased the feeling of helplessness and exclusion. It was particularly stressful when partners did not receive any information about the women’s current condition for a longer period. One interviewee described how he was left without any information for several hours after his wife was finally admitted to the hospital. For data protection, none of the HCW he called could address his growing despair from feeling totally excluded. Finally, his wife, who had collapsed in between of pain, could call him and he was allowed in.

One hospital provided a limited number of family rooms where fathers were allowed to stay with the mother and newborn child. However, they were not permitted to leave the room during their stay, neither for a walk nor for a cup of coffee. While the postpartum women were physically weakened and the ‘room confinement’ was okay for them, it was reported as a burden for the partners (Table A1). 

### 3.3. How Could the Perceived Burdens Be Managed?

The awareness of an exceptional situation ran like a common thread through nearly all interviews. It became clear that the restrictive measures were seen as necessary in ensuring safety for the baby and the adults. Many efforts to compensate for the lack of personal face-to-face contact were undertaken. As most participants were well experienced in digital communication, they used it frequently to stay in touch. 

The support perceived from staff was repeatedly stressed as a factual and emotional facilitator. The time span between arrival at the hospital and the transmission to the delivery room was reported as the major phase of feeling alone. However, some respondents described how midwives tried to comfort them by buffering restrictions. In the maternity ward, they felt that HCW tried to make up for the absence of visitors and the overall straining situation. In addition, while pandemic regulations were strictly applied, a thoughtful exception was made for a family with a disabled child when the father was allowed to join the maternity ward. 

Finally, all discharges were joined and facilitated by the hospital staff, who also enjoyed the reunion of the family (Table A1). 

### 3.4. How Did a COVID-19 Diagnosis Challenge a Birth?

Two women in the sample had tested positive for COVID-19 after admission. One of them previously had mild symptoms of infection, but a negative test result some days before. The other woman had not felt any symptoms at all. Accordingly, both were surprised by the test result, which was only available after a few hours’ stay in the hospital. The birth dates of their babies were January and May 2021, respectively, in different hospitals. In January 2021, COVID-19 tests and sufficient hygiene equipment were already available, but staff were largely unvaccinated, whereas in May the hospital staff were fully vaccinated. Moreover, the incidence rates in May were significantly lower than in January. Differences between the two narratives are presented in detail as they may display the challenges according to these phases. 

They were satisfied with the care until the very moment when the test results were released. From then on, everything changed, as they were transferred to separation rooms, cared for by fully covered HCW, and most of the time left on their own. Both interviewees were provided with either a baby monitor or a phone to call since staff members could not spontaneously check on the women. However, the phone connections did not work properly for different reasons. The feeling of being isolated was increased by the masking of the caregivers, who were completely hidden behind hygiene protection material. 

Both women experienced a delayed first stage of birth. For the woman who gave birth in May, the lack of social contacts and support was described as dominant. She perceived it as the cause for her being in labor for 24 h without significant progress in the opening of the cervix. During that time, she had to stay in the separation room without a chance to walk around or take a warm bath. For the woman who gave birth in January 2021, the birth turned into a real nightmare, starting with an unbearable level of pain. Due to the lack of a working phone connection, she had to scream for help. 

A major difference between the two women was the participation respective exclusion of the partner during the birth. While the already fully vaccinated partner of the woman who delivered in May enforced his presence, there was no exception given in January. These conditions led to the facilitation respective exacerbation of the already burdening situation. After she had given birth, the woman who was joined by her partner was sent to her former separation room in the delivery ward. She was satisfied with the postnatal care, even though she had to accept longer waiting times due to measures such as putting on infection-protective clothes. 

In contrast, for the woman who gave birth in January, the nightmare continued during and after the delivery. After two days of severe labor pains, a C-section had to be performed for medical reasons. By then, the woman was already completely exhausted from the pain experience, isolation, and feeling out of control. She reported crying and not being able to welcome her baby happily. After the C-section, she was referred to a ward provided for patients with COVID-19 infections—independent of obstetrics. As she still had not received fresh clothing and bedding for the baby after several hours, she had to explicitly demand it. Again, she was left socially isolated. The care was perceived by the mother as inadequate in the following days as staff avoided entering the room due to a risk of infection. However, at least the woman felt supported by doctors and midwives who came from the maternity ward, one of whom provided her with a crib and diapers after the end of her regular shift. Due to the lack of breastfeeding support in the hospital and the absence of an outpatient midwife, she could not solve her breastfeeding problems and finally switched to bottle feeding. In hindsight, she recalls her despair and self-doubt. The interview was conducted three months after the birth, but the participant still felt traumatized by her birth experience and at that time she could not imagine having another child (Table A1).

### 3.5. Obstetric HCW Interviews 

In this section, professional perspectives are reported from interview data. Again, the topics answer the main research questions and significant quotes are presented in Table A2.

### 3.6. What Adjustments Did the COVID-19 Pandemic Containment Rules Require?

The obstetric staff had to rapidly adapt their clinical processes and structures. Regarding organizational aspects, several rules were implemented such as wearing face masks, providing separated rooms, enforced hand hygiene, and routine testing of the admitted patients. Against the situation of general uncertainty, the staff members also perceived a lot of fear and uneasiness in their professional and private lives (Table A2). 

#### 3.6.1. How Was the Care for the Women Affected?

The containment rules required not only wearing face masks but maintaining physical distancing when possible. Verbal communication became more important due to face masks. Moreover, women stayed longer in the maternity ward, whose staff described the difficulties of getting in touch with women, verbally and physically. The situation was even harder if language barriers existed and nurses had to be careful not to cause confusion. The exclusion of men was also reported as a tangible burden that HCW could hardly compensate for. However, they used several strategies to overcome physical distancing by speaking more clearly, gesturing, or showing the body parts they had to examine. In case of emotional needs, they sometimes crossed the hygienic rules and sat next to a woman or even gave her a short hug. 

On the positive side, joining the women when they reunited as a family at discharge was also reported as a very pleasant moment for the staff (Table A2). 

#### 3.6.2. Perception of the No-Visit Policy on the Ward

In broad agreement, the interviewees considered the exclusion of visitors as a positive experience for mother and child, directly observable in the improved breastfeeding success. This consideration is in line with the reported perception of the mothers. The impression was given that some women found the explicit ban on visits helpful, as they did not have to ask for privacy themselves. However, the decision not to allow fathers and siblings in was pitied, as in case of longer stays, the partners were missed as emotional support and sometimes also as a backup for the staff in challenging situations. On the other hand, the no-visitation rules led some women to ask for an early discharge, which was considered medically critical by some professionals. The positive experience with the exclusion of visitors other than the father led many respondents to the consideration that some of the restrictions would be maintained after the pandemic (Table A2). 

### 3.7. How Were Professional Obstetric Standards Affected?

The COVID-19 restrictions reset some essential standards in obstetrics. The exclusion of accompanying persons from the maternity ward was perceived as a step backwards to the times when women in labor were at the mercy of their professional caregivers and HCW raised concerns that this was a step back from their professional standards. Restricting partners to the last phase of birth in the delivery room was also a professional dilemma. In the face of staff shortages and additional tasks due to compliance with COVID-19 rules, the lack of personal support could not be compensated for by the staff. Neither could they devote the time nor replace the familiarity and intimacy of the next of kin. In addition, the staff lacked a mediating and helping person in decision making and cheering up the mother. The ban on partners entering the hospital for several weeks contradicted the professional self-image and was sometimes very stressful when HCW had to restrict the access of desperate partners. 

However, the criticism was weighed against an outbreak of infections that would have been catastrophic for pregnant women who depended on delivering at a university hospital for medical risk factors. Against this background, the restrictive measures were seen as partly appropriate (Table A2). 

### 3.8. How Did the Mothers (and Partners) Cope with the Restrictions from the Professionals’ Point of View?

Against the background of the pandemic, the women giving birth and their partners had largely come to terms with the containment measures. At the beginning of the pandemic, more explanations were needed to reach compliance. Some partners’ reactions were described as harsh, although the staff members reported having empathy for emotional reactions. In contrast, arguments about partners wearing face masks or complying with other basic rules was reported as frustrating as staff are required wear masks all day long. 

On the positive side, some (social as well as digital) strategies were noticed that succeeded in easing the situation, e.g., chatting online or having lunch with the partner outside the hospital in warm seasons (Table A2). 

### 3.9. How Could the Birth of COVID-19 Infected Mothers Be Managed?

For obstetric staff, caring for women infected with COVID-19 posed an additional organizational and professional challenge, even though severely ill women were cared for in a special ward. To reduce the risk of infection for the staff, (becoming) mothers, and other patients throughout the hospital, separate rooms were set up and special care procedures were adjusted. Ante- and postnatal care in the delivery ward was provided in separate rooms. The women were provided with phones to reduce the frequency of physical entries into their rooms and women were expected to voice their needs verbally and without face-to-face contact. Their rooms were entered less frequently than usual to not only reduce the risk of infection but also to minimize the time needed to put on protective clothing.

In accordance with the interviewed COVID-19-positive women, it was particularly difficult to face the consequences of an unexpected test result. Suddenly, women were only cared for by staff in full protective gear which put an additional burden on the women, who were already worried about a negative outcome for their baby. Furthermore, undertaking a delivery was described as extremely demanding, as the women often struggled to breathe as they had to wear face masks all the time. In addition, the protective equipment provided a distance that could hardly be moderated by verbalizing and gesturing. Even if the professionals tried to deliver the best possible care under the given conditions, they could not live up to their usual standard. 

In the maternity ward, the situation for the women was perceived as very stressful, because they were all socially isolated. The rooms with infected women were entered in a lower frequency, but HCW mostly stayed a bit longer and completed several tasks in one visit to save time for putting on the protective materials. 

Concerning the professional standards, HCW are faced with the dilemma to protect patients and at the same time care sufficiently for the infected women. Before the vaccines were available, the staff were extremely worried about becoming infected, as they would have placed other patients in great danger. When weighing the risks, COVID-19-positive women had to undergo a stressful and isolated birth, even though the staff tried to support them. The situation was straining and demanding for the staff, too. One physician was still burdened by the experience of a mother dying of COVID-19 after an emergency C-section that rescued her child (Table A2).

### 3.10. What Were Resilience and Weakness Factors of Obstetric Health Care under COVID-19 Conditions?

The data above display the significant effort of the HCW to provide the best possible, safe care under the conditions of the COVID-19 pandemic. The rapid adaption to organizational challenges such as availability of separated spaces and restructuring of tasks including frequent adjustments is proof of the resilience of obstetric care. This worked by partly putting aside personal needs and responding flexibly to the new situation. However, despite all efforts, HCW could not live up to their usual professional standards, as women in labor were not compensated for the absence of partners or other accompanying persons. This contradicts the professional standard, in which the partner is seen as part of the family instead of a ‘foreign’ visitor. Furthermore, obstetrics could not replace partners with professional support during the phase before birth, when many women felt left alone. More staff, e.g., interns, could have been helpful at this time. As a consequence, women lacked an important source of emotional but also organizational support. Even though the interviewees complied in general with the containment rules, many of them reported burdening experiences besides perceived support and care. For the partners, exclusion from the hospital and a lack of information access resulted in helplessness. 

Taken together, the three different parties mentioned many overlapping and some distinct difficulties. Helplessness and dealing with uncertainties were common topics, such as social isolation—either feeling socially excluded (partners) or having to exclude others (HCW). Limitations in communication was perceived by all parties with partial compensation options and bridging functions by means of digital devices (phone calls or video calls). Figure 1 provides an overview of the perceived barriers and resources. 

On the positive side stands the undistracted bonding, which was reported from mothers and professional participants. This unexpected positive outcome meets the long-term perception of the staff that especially large or demanding groups of visitors can pose negative effects on the mother and newborn. Furthermore, the containment rules contributed to safe care, which was also seen as supportive from the mothers’ and partners’ side. 

In conclusion, managing and dealing with the containment rules was partly seen as a burden that both maternal and professional participants had to cope with. 

## 4. Discussion

The COVID-19 pandemic has challenged individuals and society, but foremost the health care system. At the same time, obstetric HCW are faced with organizational and emotional challenges: The COVID-19 pandemic is putting standards of professional care at risk [31]. In other fields, standard procedures can be adapted and reduced to protect resources [18,32]. In this context, obstetrics differs from other medical fields: The birthing process cannot be “put on hold” as pregnant women must give birth during pandemic circumstances. Obstetric ‘patients’ are usually healthy young women with a high degree of self-determination and active participation in the birthing process, supported by their partners. It is unquestionable that satisfaction with the birth experience and subsequent bonding with the newborn are affected. This may also influence the entire relationship within the family and subsequent decision-making processes such as becoming pregnant again or deciding for a C-section [33]. 

This study aimed to examine the challenges and restrictions that mothers, their partners, and obstetric staff had experienced due to the COVID-19 pandemic through qualitative interviews. A focus rested on the perceived weaknesses and resilience of the obstetric health care system in managing the pandemic. Against this background, the subjective perceptions were exploited from mothers and obstetric HCW from their respective views on birthing under the COVID-19 restrictions to answer and discuss the research questions in the following. 

### 4.1. How Do the COVID-19 Restriction Rules Affect all Phases of Pregnancy from Mothers and Professionals’ Perspectives? 

The impact of COVID-19 restrictions appears significant for both patients and staff. Since pregnant women and their unborn children represent a particularly vulnerable group at risk of infection, containment measures are heavily emphasized [34]. The interviewed personnel described their involvement in applying the frequently adapted safety rules as extremely demanding, organizationally, socially, and individually. For them, most of the work processes have changed or are more complex, which significantly affected their daily workload. Many mothers emphasized how the anticipation of containment rules in the hospital facilitated proactively coping with them. In contrast, not all interviewed professionals agreed with the full extent of all measures. However, they have an active role in asserting them, while mothers must endure them rather passively. 

The results match prior qualitative research that “COVID-19 changed the whole experience”. In a recent study from England, the researchers reported that most women were able to have positive birth experiences even if the situation was affected by COVID-19, indicating resilience [35]. Nevertheless, patients have difficulties dealing with objective isolation and feeling isolated in a vulnerable situation [35]. Especially when experiencing health care discrimination due to COVID-19, levels of postpartum stress and mental health issues are increased [36].

### 4.2. How Is the Role of the Partners and the Professional Obstetrics Standard Impacted? 

The father’s involvement during pregnancy and birth, at least in Germany, has become a central part of the clinical birth process during the last few years. The professional conception followed a social change in which women had fought for fundamental self-determination in all gender-related issues of reproductive health [14]. A central theme nowadays (i.e., until the beginning of the pandemic) is the active participation of the partner in the birthing process: The conception of the partner is not only to support the mother but also to take on the role of fatherhood and to be a reliable bonding partner for the newborn. 

With the onset of the pandemic, fathers or other accompanying persons are at risk of being excluded from antenatal check-ups and the hospital, except for the second stage in the delivery room. Typically, they approach the hospital, but against the current conception of the partner, partners can only bring the women to the door and then must wait to be let in. Instead of providing emotional and physical support to women in labor, partners spend the waiting time at home or in heat/cold outside the hospital without any infrastructure. This situation challenges the role of the caring partner and at the same time contradicts the professional standard of obstetric care, in which the father plays a central role. All interviewed professionals mentioned how negatively they were affected by the absence or only brief presence of the partner since the reliable supporter and mediator of the woman was missing. Prohibiting partners and support during labor has previously been identified as a domain of obstetric with the potential for severe consequences on the birth and thus the emotional and physical well-being of both mothers and infants after birth [37].

These restrictions lead to the fact that accompanying persons were not only kept away from the women during a period of great vulnerability but also from active participation at the beginning of parenthood [14]. This is partly compensated for by using digital contact options. However, skin-to-skin contact is crucial for the father–child attachment, and long-term consequences of the lack of father–child bonding during the first days due to the COVID-19 pandemic are largely unknown until now [38].

From a professional viewpoint, keeping partners away from the women is perceived as professionally and emotionally problematic. In addition, staff interviewed as part of the study are concerned about a significant number of women with complex medical conditions who decide to deliver in hospitals with fewer restrictions: This harbors the risk that the standard of care is below what at-risk women require. These findings are in line with a recently published study that found that obstetric staff were morally distressed in having to balance the risks of COVID-19 versus the need to care for mothers [39].

### 4.3. How Are the Perceived Burdens Managed?

The staff reported difficulties in explaining and enforcing the containment rules during the beginning of the pandemic because mothers and partners did not always agree. Harsh reactions and bargaining were also reported, although they were described as partly understandable. With the proceeding of the pandemic, limitations are perceived as mostly tolerated. One woman even emphasized explicitly how comfortable it was for her to feel in a safe place.

Several strategies to compensate for the contact limitations have been described from both sides [17]. As women giving birth belong to a group of patients characterized by high technology affinity, they are familiar with all kinds of social media. Frequent digital communication was highlighted as helpful to make up for the lack of face-to-face contact with partners, family members, and friends. When weather and physical conditions were adequate, the women bypassed their partner’s exclusion by spending time outside the hospital with them. Furthermore, attention and encouragement from the staff were perceived as emotionally supportive. In the case of a baby born with a handicap, it was enforced that the father could stay in a family room even though it was not permitted in general. Nurses from the maternity ward reported how they tried to back up the women when they seemed miserable. Some women even gave the impression to have grown with their self-esteem as noticed by one nurse. The reuniting of the family at discharge was also stated as a great pleasure for the maternity ward staff, who escorted mothers and newborns to the door. 

### 4.4. How Is a COVID-19 Diagnosis Challenging the Birth from Women’s and Staff Perspectives?

Women who tested positive for COVID-19 faced the worst possible exclusion and restrictions which could be described as a nightmare, from both professional and patient sides, respectively. The risk of infection transmission leads to drastic restrictions. Births under these conditions fall far behind the usual obstetric standards. Although it is obvious that the women feel vulnerable, anxious, and lonely, they are locked in small rooms that staff entered as infrequently as possible. The immense burden is presented in detail by mothers and confirmed by HCW. The much greater burden on the mother who gave birth in January may be explained by the lack of vaccines at that time, which placed the HCW at great risk of being infected, too. Therefore, one positive finding was an improvement which was noticeable from the earlier birthing to the latter one with restrictions but which was somehow more tolerable.

Giving birth when the staff and partner are already fully vaccinated is an improvement to the birthing process in an unvaccinated environment. However, even in the latter environment, the partner struggled for several hours until he was permitted at the delivery room. Being without their partner, women may experience a nightmare of extended loneliness, extreme labor pain, an emergency C-section, and the feeling of not being properly cared for. Considering this experience, young mothers may even decide (such as a participant in this study) against having a second child. This was found in previous studies: A higher perceived health discrimination followed by birth-related PTSD was found for pre- and peak-pandemic times [36]. 

Although these women have very different experiences, their emotions and recollections represent many women suffering from giving birth while being COVID-19 positive. In a recent study, the authors found that approximately 50% of COVID-positive women experienced higher levels of pain during birth, acute traumatic stress responses, and a violation of their rights [37,40]. Due to the risk of traumatic birth experiences and subsequent psychological morbidity, more awareness and effort, resources, and evidence-based effective guidelines are needed. Significant distress such as traumatic birth experiences can potentially cause long-term harm; therefore, trauma-informed interventions and support are needed [41].

Typically, HCW try to care for the women as best as possible under the current environmental circumstances. They reflect and commiserate with the extremely stressful and negative situation for the women involved, but at the same time have in mind that other mothers—and patients in the entire hospital—need to be sustainably protected from infection. Even though they tried to care for the women as best they could, they fell far short of their usual standards. This, in turn, may cause a professional dilemma and also impact stress levels. Protective clothing, including face masks, makes communication and interaction difficult. The time-consuming preparation for women’s visits results in infrequent visits, which match the hygiene concept. In this respect, they find themselves in a professional conflict that cannot easily be resolved. This conflict also exists in the literature. Two recently published studies from the UK and Italy showed higher psychological distress levels impacted by COVID-19 in obstetric HCW [42,43].

The negative experience needs to be considered especially in the face of the lack of protective measures, which means that there was a higher risk of infection for the staff and thus for other patients, too. In the meantime, current regulations have been adapted for the increased protection of staff and other patients by vaccination. Although vaccination seems safe for pregnant women, some are still skeptical and hesitant to get vaccinated and vaccination recommendations need to be promoted [44,45].

### 4.5. Resilience and Weakness of the Obstetric Health Care under the COVID-19 Conditions

The need for adaptions as well as organizational flexibility and the determination required to manage the challenges brought by the COVID-19 pandemic is proof of the resilience of the obstetrics departments. This can also have implications for reflections on alternative ways of working when facing the COVID-19 pandemic. The challenges require a significant effort from HCW who performed very hard work, organizationally and mentally. The data show how they focused on providing the best care to mothers, even though they personally faced uncertainties and restrictions, as did the population. It was possible by means of a high commitment to safety measures, compliance with care regulations, and the general Infection Protection Act. 

At the same time, close cooperation within the team is required, especially if staff are torn apart by adapted care models and have to compensate for left-out accompanying persons. In their daily work, professionals cope with restrictions that put their activities behind their professional standards to comply with containment rules [39]. This appears as a major problem when infections can only be prevented through contact avoidance. However, many more alternative ways of working when facing the COVID-19 pandemic are now identified such as safety gear (FFP2 masks, etc.) and vaccinations, but also more efficient communication techniques using technology. For instance, the use of digital tools to involve partners in the treatment process or online information events for parents-to-be along with converting staff meetings to an online format and structuring handoffs is recommended even if time is limited or staff cannot be in the same room. In addition, HCW tested frequently and refrained from joining breaks together or business trips. 

At this time, political and clinical leaders were and are still aiming at not becoming a hotspot for contagions. This is definitely the only correct procedure as this would have been a major disaster for obstetric care in these regions, and in general: If a university hospital would have had to shut down, women with risk factors might not have been able to receive the highest level of care and even more disastrous outcomes could have occurred. Risk assessment and negotiations remain a daily routine with no alternative ways of working when facing the COVID-19 pandemic. However, communication is key in this process, and many options for improvements to prevent and handle anxiety, uncertainty, stress, and crisis.

In weighing the decisions, the partners were treated as visitors at best and not as fully responsible birth partners, as many HCW classified them. In such cases, women in labor lack their main resource of emotional and instrumental support, which is particularly important in overcoming communication barriers due to the environmental measures in place to prevent the spread of COVID-19 infections. Against this background, the visitation policies harbor the risk of adverse effects when women with high-risk pregnancies preferred hospitals with less rigorous rules or requested early discharges. With this in mind, obstetric HCW are able to enforce the presence of the father at least for the last phase of birth [46,47]. However, if this is not the case, in line with international studies, a concern remains about the restrictions: They may affect ‘the rights of birthing women on safe motherhood during the COVID-19 pandemic’ or fear of a ‘fallback to alienation during intervention-focused hospital deliveries’ if partners are excluded from some part of the process [48,49]. Thus, alternative ways of working when facing the COVID-19 pandemic require awareness but also skills such as speaking up and shared decision making [50].

Managing a pandemic is further exacerbated by existing weaknesses in the obstetrics departments which have become more obvious over the course of the pandemic. This mainly refers to staff shortages, frequent staff turnover, burdening administrative tasks for the study hospitals, and the acute lack of space and other environmental restraints [50]. These circumstances increase the stress of the mothers giving birth, as the absence of partners or other familiar persons may not be compensated for by HCW or digital communication. The lack of space results in additional organizational challenges, as the physical separation between women with confirmed COVID-19, those with pending test results and women who test negative are required. 

As the major weaknesses refer to the quality of the health care system, some adjustments could be made on the department level to address the physical and emotional safety of the mother, baby, and partner. The WHO strongly recommends that all women, even with suspected or confirmed COVID-19 infection, have a ‘companion of choice’ by their side during labor and childbirth because of the benefit for both mother and child from the presence of a familiar person [45,51,52]. This includes acknowledging the father as a partner involved in the birth process, who should be present from the beginning of the labor phase if sufficient protective measures can be ensured [51,53]. Further organizational adjustments appear feasible: Given the need for data protection, personal codes, or other technical means for information, access by phone could be agreed upon for not cutting off relatives from the current state for too long. Moreover, matching previous international studies, the finding from this study also indicates that many HCW experience burnout due to the pandemic. To protect them from long-term stress or leaving the job, psycho-social services are needed to prevent frustration and emotional exhaustion [54,55].

More research is needed, especially on long-term effects. This study explored the perception of the participants and did not cover long-term consequences or objective repercussions. However, taking into account such long-term, secondary effects might be key also from a societal level; examples of such are rational decisions to not have further children, unconscious etiology of infertility or costs due to necessary follow-up treatments or psychotherapy to overcome PTSD, depression, or anxiety. This requires further comprehensive research, even though initial studies have indicated that facing the challenges of COVID-19 during childbirth has negative long-term effects on the (mental) health and well-being of mothers and their children, especially in the case of experiencing a birth trauma due to a positive COVID-19 test result [40]. Obviously, for ensuring safe birth conditions and meeting the needs of all included parties, a health care system is needed that considers the lifelong impact of a safe birth with all its risks and secondary consequences [37,48].

### 4.6. Limitations

Several limitations must be considered. Qualitative analysis is subjective by nature and interview bias cannot be completely excluded. As such, it is possible that findings may reflect the personal biases of the investigators. Furthermore, our findings present characteristics of two German university obstetrics departments, where the strictest COVID-19 rules were applied. Therefore, our findings can only be partially generalized to other hospitals or birthing environments. 

The interviews were conducted between February and August 2021 and the results refer to the perception in this period of the pandemic. They addressed challenges and consequences relating to the COVID-19 pandemic during the birth process. Thus, this study did not reflect on any influences the pandemic might have had on birth preparation as well as the postpartum period after discharge. This would be another research question also worth investigating in the future.

In addition, the study was conducted within a research project on safe communication in obstetrics. Most of the maternal participants (and some of their partners) had participated in communication classes about safe communication before the birth. This might have raised their awareness of personal needs during the interviews and hence caused overreporting and special attention to potential challenges.

## 5. Conclusions

Our study shows that a pandemic and especially the associated containment measures including contact restrictions have a substantial and largely negative impact on childbirth experiences. Especially at the beginning of labor, women feel left alone or even socially isolated. At the same time, the perception of insecurity, which could not fully be compensated for by staff members or technology, is dangerous. Accompanying persons are greatly limited in the time they can support women in labor. They are thus not able to be a companion throughout the whole process, even if women strongly needed them. 

Perceived burdens are partly managed by compensation strategies such as digital communication and emotional support from obstetric staff members. During the postpartum stay, most women and staff members appreciate the no-visitation rules. Nevertheless, a COVID-19 diagnosis is completely detrimental to the birth process in one of the cases that left the woman overwhelmed with a traumatic birth experience. The containment measures require a high number of personal and organizational adaptations as well as financial investment. This may be challenging for both patients and staff members who fall beyond their usual standard of care and suffer from the experience. 

Going forward, the provision of guidelines and resources to support staff to care for pregnant women during labor and childbirth will greatly enable the supply of higher quality care and experience for birthing women. The fathers and other birth companions need to be part of the process which can only be realized if effective containment measures are in place leaving enough room for adequate care. More work is needed to support HCW to maintain or recover their resilience. The exclusion of partners should only be limited to such instances where a clear risk level outweighs the benefits of supporting laboring women and bonding with their newborns. 

## Figures and Tables

**Figure 1 ijerph-19-01486-f001:**
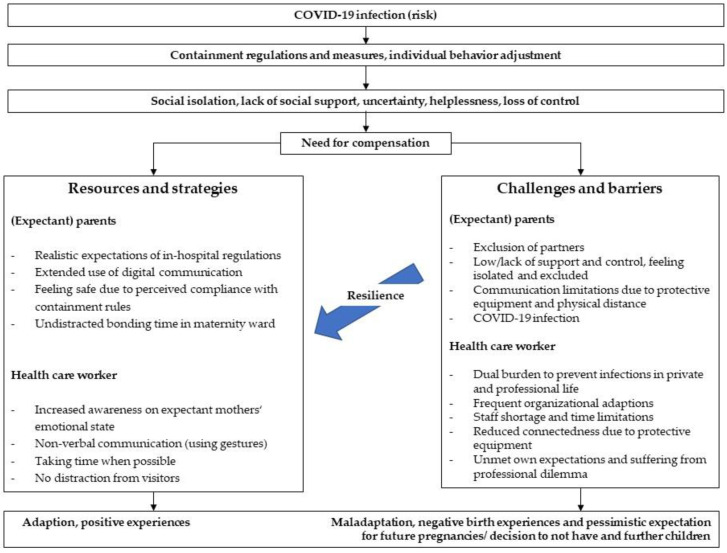
Overview on perceived barriers and resources.

**Table 1 ijerph-19-01486-t001:** Demographics of participating mothers *.

Characteristics	Distribution in the Sample of Mothers
Age at birth	22–46 years, Median 34 years, mean 33 years (SD = 5)
Migration Status	5 (20%) of which: 3 (12%) < 10 years immigrated themselves 2 (8%) second generation
Parity	19 (76%) first-time mothers 6 (24%) second and multiple births
Birth mode	18 (72) vaginal 7 (28%) caesarean section
Twin birth	1 (4%)
Child with disability	1 (4%)
COVID-19 tested positive	2 (8%)
Interview duration	19–95 min Median 35, mean 38 min; (SD = 15)
Interview format	22 (88%) via TEAMS with camera on 3 (12%) by telephone/TEAMS camera off
Interview period	February to July 2021
Period of birth	July 2020 to May 2021

* Since partners complemented the interviews, no demographic data were obtained.

**Table 2 ijerph-19-01486-t002:** Demographics of participating HCWs.

Characteristics	Distribution in the Sample of HCW
Occupational Age	2–30 years Median 6 years, mean 12 years (SD = 7)
Gender	Female, *n* = 9, male *n* = 1
Profession	Midwives *n* = 4 Doctors *n* = 2 Nurses *n* = 2 Assistant *n* = 1
Professional level	Superior *n* = 3
Interview duration	17–75 min Median 32, mean 35 min (SD = 16)
Interview format	9 (90%) in person in the clinic 1 (10%) via TEAMS camera on
Interview period	July to August 2021

**Table 3 ijerph-19-01486-t003:** Interview guide with participating mothers and partner.

Do You Remember Getting Your First Information About births in Your Life, and If Yes from Whom? Do You Recall Emotional Association to Birth Experience—What Describes the Feeling You Associate with Your Birth the Best?
How were you affected from COVID-19 during your pregnancy?
Current birth experience: How and with whom did you prepare for your birth?
Could you please describe your birthing experience as precisely a possible from the moment you decided you have to go to the hospital? Who did you get support from? How did you stay in touch with your partner? If a partner participated: How did you feel after you left your partner at the hospital door? When was the partner allowed in? If a partner participated: When were you allowed in? How did you spend the meantime?
How was the communication affected by the COVID containment measures?
If you stayed at the maternity ward, what was the care there like?
Overall, what was good? What did you miss?
What could have been better? Who should have done something differently?

Participating partner completed or answered questions from their perspective.

**Table 4 ijerph-19-01486-t004:** Interview guide obstetric health care workers (HCW).

How Long Have You Been Working in Obstetrics?
What major challenges have you had to overcome so far?
Do you remember the first time you became aware that a pandemic was emerging?
What organizational changes were implemented at your facility? What level made the decisions? (Hospital leadership, birth center leadership, group autonomously?) Did you have sufficient protective materials? How were the structural adjustments communicated?
How did you perceive the reaction of the women? Resistance, reactions, acceptance, questions? Were there changes over time? What feelings did you perceive among the women? Fear, insecurity, feelings of being left alone or socially isolated?
(How) have you tried to compensate for the absence of partners? (Staying in the delivery room, more caring, reassurance). What was your experience with limited partner access?
How did you perceive the impact in the maternity ward? (No support from partner/‘room arrest’ partner, no other visitors)
Are there any changes that you would like to see to be continued after the pandemic?

**Table 5 ijerph-19-01486-t005:** Aggregated findings: Impact of COVID-19 pandemic on varying target groups in different phases of birth.

Target Group	During the First Phase of Labor (Period Before Admission to the Delivery Room)	In the Delivery Room	In the Maternity Ward
(Becoming) mothers	Felt socially isolated, anxious, unsafe, and uneasy	Attendance of partner in delivery room huge relief	On their own from the time they left delivery room
	Suffered from being without a partners’ company	Compensation of limitation was not completely possible	No external visitors allowed, thus mothers relied solely on assistance of nurses and midwives
	Were only routinely monitored by midwives	Digital tool partially used	Coped well with social isolation but some reported missing partners
	Uneasiness exacerbated due to uncertainty of time the partner would be allowed in	Left in delivery room for several hours after birth before the father had to leave: highly valued and even considered a personal privilege	Appreciation of not having other external visitors because of calmer atmosphere Suffered without a familiar person, esp. if not fluent in German
	Communication with staff was hindered	Communication with staff was hindered	Wanted to be discharged as soon as possible, even if obstetricians supposed it was too early
HCW (obstetricians, midwives, and nurses)	Awareness that women needed more attention, but lacked time to act accordingly	Were—more or less—as present as before the pandemic	Emphasized appreciation not having other external visitors
	Felt sympathy for women	Communication with women in labor and the accompanying partner was complicated and alternative options were not possible	Calmer atmosphere and positive effect of rest and exclusive mother-infant togetherness on successful breastfeeding and bonding
	In case of sufficient resources, still regretted not replacing a close partner or another next of kin	Interprofessional conflict with handling the situation	Tried to compensate that mothers were socially isolated by spending more time
		Making effort for mothers was straining	Only felt partially that their effort helped
Partners	Were/felt left out	Communication with staff complicated	Left in delivery room for several hours after birth: highly valued and considered a privilege
	Felt isolated helpless	Compensation of limitations not completely possible	Had to leave early and felt excluded

## Data Availability

Original data are not available for data protection reasons.

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
