# Peer review of "Birthing under the Condition of the COVID-19 Pandemic in Germany: Interviews with Mothers, Partners, and Obstetric Health Care Workers"

_ijerph, 2022, doi:10.3390/ijerph19031486_

Round 1

Reviewer 1 Report

Dear Editor, I hope you have enjoyed these holidays. First of all, thank you for the opportunity to contribute to this fundamental issue during the pandemic. Second, although it is an extensive investigation, it lacks investigative methodology to support the great work that has been done. Finally, it is suggested to improve according to the comments.

General features:

Title: although it is clear and reflects the research, it is suggested to reduce its extension.

Abstract: it does not clearly state the objective, nor does it mention the paradigm, focus of the qualitative research, or scope of the type of study. Nor does it clarify the research participants, results from analysis technique, and ethical aspects.

MeSH: do not conform to MESH. Check according to the website.

Introduction: I believe that the first and second paragraphs should be improved because they do not address the object of the study's problem. The main actors in the matter at the global level, such as WHO, should be mentioned.

Methodology: does not mention the paradigm, focus of the qualitative research, design, or scope of the study. Nor does it make clear the research participants. Regarding the instruments, it does not mention how the construction of the interviews was carried out if they were applied to verify the questions they responded to the object of study. The collection of information does not mention how the ethical principles, duration, and dates of application were respected. The rigorous criteria are not specified, although authors describe triangulation.

Results: check to number as there is one before the title. There is a number at the end of each narrative; its origin in methodology should be explained. It is unclear whether these are emerging or previous categories and whether subcategories or units of meaning are derived from each. Your presentation should be enhanced with a brief explanation in the initial presentation of the results and with numbering to aid understanding. For example, the verbatim in row 192 is insufficient to support the above category. In the category that starts at row 284, subcategories or topics are presented; there are many narratives with brief explanations each. Improve overall presentation.

Discussion: the presence of a table stands out in the discussion, its use must be justified with reference according to the paradigm and research focus used.

References: the style of references needs to be improved; some only present the link. However, they are updated with 89% of the last five years, the absence of key actors in the subject such as WHO stands out.

Author Response

Thank you very much for your feedback. Please see in the attached file how we dealt with your suggestions.

Reviewer 2 Report

  1. The title "Interviews with mothers, partners and...", you use partners interchangeably with fathers and husbands, Is it a common way of saying?
  2. Line 42 "Infektionsschutzgesetz"
  3. The title of table and figure, delete the . at the end
  4. Table 4, all the N, change to n
  5. Line 227 May 2021
  6. Line 382 delete an extra ,
  7. Line 797 birth partner, is it same as partner that the authors mention throughout the whole article?
  8. In general, negative effect of social restrictions on childbirth experiences was presented strongly. However, little was mentioned the balance between mother's feelings, safety and staff worker's burnout. What is the alternative ways of doing when facing COVID-19 pandemic? Or is there an alternative way? This paper is long, other than quote what was said verbatimly. The key themes need to be integrated and presented clearly.

Author Response

(The authors gave the same response as above.)

Reviewer 3 Report

A very interesting and worthwhile paper on mothers' and HCWs' experience in birthing during COVID-19. This paper shines a light on obstetrics during a pandemic (a relatively new and unknown circumstance in modern society). The methodology is sound and provides insights from mothers, partners, and HCWs, and the findings make complete sense.

This is a very well researched and written paper. However, there are numerous minor issues with spelling, phrasing, referencing, etc.

  • Line 57-60: is there reference to go with these statements?
  • Line 65: insert comma after delivery
  • Line 67: "health care worker" should be a plural
  • Line 68-69: "But they were not only double burdened." this should not be a freestanding sentence. Perhaps combine with the following sentence, e.g., They were not only double burdened but, rather, the demands...
  • Line 69-72: is there a reference to go with this statement?
  • Line 74: "containing" should this be containment?
  • Line 78-81: reference? (or if this is referring to the previous citation, please make it more clear)
  • Line 109: numbering of all subsequent headings are wrong. They all say 1.
  • Line 113: "Detailed information is been described" should either be "is described" or "has been described"
  • Line 128: "All interviewees had previous participated" should be "previously"
  • Line 136: Table 2 and Table 4 title should be "Demographics of..."
  • Line 139: no apostrophe after patients
  • Line 141: "accomplished according..." is this a typo. I'm not sure what is being said here. Please rephrase.
  • Line 152: "All participants gave written informed consent..." this sentence would probably be better somewhere in the previous section.
  • Line 199-200: "the partner at their side" should be "their partner by their side"
  • Line 227: "Mai 2021" typo
  • Line 228: "complied with" should either be "complied with them" or just "complied"
  • Line 236: "Beside of..." should either be "besides", "apart from", "despite", "other than"...
  • Line 255: "In case the" should be "In the case where..."
  • Line 288-289: Please rephrase. Avoid equating young age with digital literacy
  • Line 352: there was only 1 woman, right? "women" should be "woman"
  • Line 368: "accepted" should be "accept" and "due to measures as" should be "due to measures such as"
  • Line 532: "basic rules as" should be "basic rules such as". Also, should "caused discussions" be "caused arguments"?
  • Line 569: "Undertaken" should be "Undertaking"
  • Line 577: this is the first instance of using "HCP". Either state it in full, followed by the abbreviation in parentheses, or consistently use "HCW" throughout the paper
  • Line 602: "bridging functions my means" should be "bridging functions by means"
  • Line 608: "were faced organizational" should be "were faced with organizational"
  • Table 5: there appears to be an empty column. Please remove
  • Line 646: "situations was" should either be "situation was" or "situations were"
  • Line 694-695: Is there a reference to go with this statement?
  • Line 699: "perceived as emotional" should be "perceived as emotionally"
  • Line 717-719: I am unclear as to what this is saying. Please rephrase.
  • Line 738: "woman" is this meant to be singular or plural?
  • Line 748: "UK" should be "the UK"
  • Line 754: "recommendations just transmit to the public". I am unclear as to what this is saying. Please rephrase.

Author Response

(The authors gave the same response as above.)

Round 2

Reviewer 1 Report

Dear authors, first of all, congratulations on the topic of research that is so relevant in times of pandemic. However, the article has been improved in several aspects, and it continues without a clear reference statement used in the qualitative research methodology that supports the great work done. Therefore, it is suggested to use authors to give internal coherence to the article.

The title has been improved.

Abstract: it does not declare a paradigm or focus of the investigation. The rest have been improved.

The introduction was improved.

Methodology: continues without mentioning paradigm, the focus of qualitative research, design, or scope of the type of study; it is not quantitative research, and these details are essential in a qualitative design. The collection of information does not mention how the ethical principles, duration, and dates of application were respected. The criteria of rigor are not specified, although triangulation by authors is described, but not what theoretical reference (the author was used).

Results: A number at the end of each narrative should explain its origin in methodology.

References: improve reference 44.

Author Response

Dear Reviewer,

thank you very much for your valuable feedback. We have revised the manuscript accordingly, and also proofread the whole paper again. We hope you are satisfied with revisions and the paper in its current form now.  

Best wishes, the authors
